# A Comparative Study on Chemical Compositions and Biological Activities of Four Amazonian Ecuador Essential Oils: *Curcuma longa* L. (Zingiberaceae), *Cymbopogon citratus* (DC.) Stapf, (Poaceae), *Ocimum campechianum* Mill. (Lamiaceae), and *Zingiber officinale* Roscoe (Zingiberaceae)

**DOI:** 10.3390/antibiotics12010177

**Published:** 2023-01-15

**Authors:** Alessandra Guerrini, Massimo Tacchini, Ilaria Chiocchio, Alessandro Grandini, Matteo Radice, Immacolata Maresca, Guglielmo Paganetto, Gianni Sacchetti

**Affiliations:** 1Pharmaceutical Biology Lab., Research Unit 7, Terra&Acqua Tech. Technopole Lab., Department of Life Sciences and Biotechnology, University of Ferrara, P.le Luciano Chiappini 2, 44123 Ferrara, Italy; 2Department of Pharmacy and Biotechnology, Alma Mater Studiorum, University of Bologna, Via Irnerio, 42, 40126 Bologna, Italy; 3Faculty of Earth Sciences, Dep. Ciencia de la Tierra, Universidad Estatal Amazónica, Km 2 ½ Via Puyo-Tena, Puyo 160150, Ecuador

**Keywords:** *Curcuma longa*, *Cymbopogon citratus*, *Ocimum campechianum*, *Zingiber officinale*, essential oils, chemical composition, volatile fraction, antioxidant activity, antimicrobial activity, mutagen-protective properties

## Abstract

Essential oils (EOs) and their vapour phase of *Curcuma longa* (Zingiberaceae), *Cymbopogon citratus* (Poaceae), *Ocimum campechianum* (Lamiaceae), and *Zingiber officinale* (Zingiberaceae) of cultivated plants grown in an Amazonian Ecuador area were chemically characterised by Gas Chromatography-Flame Ionization Detector (GC-FID), Gas Chromatography–Mass Spectrometry (GC-MS), and Head Space–Gas Chromatograph-Flame Ionization Detector–Mass Spectrometry (HS-GC-FID-MS).figure The EOs analyses led to the identification of 25 compounds for *C. longa* (99.46% of the total; ar-turmerone: 23.35%), 18 compounds for *C. citratus* (99.59% of the total; geraniol: 39.43%), 19 compounds for *O. campechianum* (96.24% of the total; eugenol: 50.97%), and 28 for *Z. officinale* (98.04% of the total; α-Zingiberene: 15.45%). The Head Space fractions (HS) revealed *C. longa* mainly characterised by limonene and 1,8-cineole (37.35%) and α-phellandrene (32.33%); *Z. officinale* and *C. citratus* showed camphene (50.39%) and cis-Isocitral (15.27%) as the most abundant compounds, respectively. *O. campechianum* EO revealed a higher amount of sesquiterpenes (10.08%), mainly characterised by E-caryophyllene (4.95%), but monoterpene fraction remained the most abundant (89.94%). The EOs were tested for antioxidant, antimicrobial, and mutagen-protective properties and compared to the *Thymus vulgaris* EO as a positive reference. *O. campechianum* EO was the most effective in all the bioactivities checked. Similar results emerged from assaying the bioactivity of the vapour phase of *O. campechianum* EO. The antioxidant and antimicrobial activity evaluation of *O. campechianum* EO were repeated through HP-TLC bioautography assay, pointing out eugenol as the lead compound for bioactivity. The mutagen-protective evaluation checked through Ames’s test properly modified evidenced a better capacity of *O. campechianum* EO compared with the other EOs, reducing the induced mutagenicity at 0.1 mg/plate. However, even with differences in efficacy, the overall results suggest important perspectives for the functional use of the four studied EOs.

## 1. Introduction

Aromatic plants have always characterised the ethnobotanical traditions and cultures of all societies in the world without distinction, finding a place as an economic botany resource in many uses including—first of all but not only—healthy preparations and therapeutic applications. Their aroma also traditionally characterises them for different and multiple uses such as cosmetics, perfuming and sanitising environments, insect repellent, and for religious and pagan propitiatory rites. The essential oils (EOs) obtained from these plants represent the maximum expression of their efficacy because steam distillation, the extraction method of choice, allows the compounds which characterise these species in terms of their aroma and multiple applications properties to be obtained in the concentrated form [1,2,3,4]. The therapeutic importance of many aromatic plants and their EOs has led to their inclusion in various pharmacopoeias which regulate their authenticity, standardisation, and application in the health and pharmaceutical fields. In addition, research into their use both as phytocomplexes and as characterising active ingredients in the treatment of various pathologies, such as infections of various aetiology, pathologies of inflammatory origin, tumours, and immune-related pathologies, is always particularly lively [5,6,7].

The evaluation of the quality and biological properties of EOs has led research to enhance their use in other areas as well, such as the veterinary sector, for the treatment, for example, of diseases characterised by decreased immunity (e.g., caused by forced weaning); that of sustainable agriculture for the identification of molecules with an insect-repellent or biopesticide action with a lower environmental impact; or that of packaging and preservatives by reducing the use of synthetic compounds [8,9,10].

Precisely, thanks to their wide use as seasoning spices, aromatic plants and their EOs have been studied—and are still being extensively investigated—for biological activities that can suggest functional applications in the food sector. In particular, modern research focuses on antimicrobial, antioxidant, and in some cases, geno-protective properties, for applications of EOs as preservatives and flavourings, protective agents against the potential carcinogenicity of substances present in foods, such as, for example, the heterocyclic amines produced by cooking, or packaging components protecting the food from microbial and chemical degradation [11,12,13].

Finally, the cultural and economic value of aromatic plants and EOs is also reflected in their inspiring role in the chemical industry of synthesis and semi-synthesis (platform chemicals) to produce compounds or new molecules of interest for various productive sectors [14].

In relation to these important premises, EOs reflect in a very effective way the physiological and ontogenetic evolution of the aromatic species of origin as well as its resilience, responding to exogenous (e.g., climate) and endogenous (e.g., hybridisation) stress factors with different qualities and quantities of secondary metabolites produced and influencing the phytochemical profile of the phytocomplex. This feature strongly affects the standardisation of the EOs produced and therefore their specific effectiveness. The objective need to use standardised EOs in terms of quality and quantity of constituents explains the need for supplies from cultivated rather than spontaneous aromatic species, to be able to control the environment and cultivation practices by minimising the variability of the composition of the phytocomplexes. Hence the need to verify the chemical composition as well as the biological properties and also to identify—with respect to specific peculiarities due to specific growth environments (e.g., latitude, climate, cultivation practices, etc.)—any new applications [15].

For several years our research group has been studying EOs from aromatic plants growing in the Amazon area, one of the largest biodiverse hot spots in the world. Starting from the ethnobotanical uses that characterise the aromatic species studied, it is of particular interest to verify the effectiveness of traditional uses from a phytochemical and bioactivity point of view, and evaluate any new applications related to the peculiarity of the chemical composition due to the specificity of the growing area, such as, for example, that Amazonian. Therefore, the present study reported the chemical composition and the biological properties of the EOs obtained from *Curcuma longa* L. (Zingiberaceae), *Cymbopogon citratus* (DC.) Stapf, (Poaceae), *Ocimum campechianum* Mill. (sin. *Ocimum micranthum* Willd.; Lamiaceae), and *Zingiber officinale* Roscoe (Zingiberaceae) with the object to valorise possible chemical and bioactivity characteristics due to the particularity of the southeastern region of Amazonian Ecuador where the plants were grown. The chemical composition of the EOs and the volatile fraction (headspace, HS) were determined, together with the evaluation and comparison of the antioxidant activity and antimicrobial properties related to identifying the compounds mainly responsible for the bioactivity. Finally, the mutagen-protective capacity of EOs was verified through the Ames test with the aim of suggesting a possible functional and protective role of essential oils in applicative and health-related fields such as, for example, the food sector.

## 2. Results and Discussion

### 2.1. Essential Oils (EOs) Extraction and Chemical Characterisation 

The essential oils (EOs) were obtained by steam distillation from fresh plant parts collected at the balsamic period from *Curcuma longa* L. (Zingiberaceae; rhizome), *Cymbopogon citratus* (DC.) Stapf (Poaceae; aerial parts), *Ocimum campechianum* Mill. (Lamiaceae; aerial parts), and *Zingiber officinale* Roscoe (Zingiberaceae; rhizome), grown in an Amazonian Ecuador area and sourced from Fundaciòn Chankuap (Macas, Morona-Santiago province, Ecuador). The EOs yields, determined on an averaged volume/dry weight basis, and their densities are reported in Table 1. *O. campechianum* gave the highest EO yield, followed by *Z. officinale*, *C. citratus*, and *C. longa.*

The chemical characterisation of the EOs performed by GC-FID and GC-MS analyses (Table 2) led to the identification of 25 compounds for *C. longa* (99.46% of the total), 18 compounds for *C. citratus* (99.59% of the total), 19 compounds for *O. campechianum* (96.24% of the total), and 28 for *Z. officinale* (98.04% of the total) through a detailed interpretation of the experimental data (fragmentation and retention indices). Sesquiterpenes were the most abundant compounds in *C. longa* EO, mainly characterised by those oxygenated (66.29%), where ar-turmerone (23.35%), α- and β- turmerone (22.81% and 15.27% respectively) were the most representative, while sesquiterpene hydrocarbons were instead consistently lower (5.32%). The monoterpene hydrocarbons (18.74%) were mainly characterised by α-phellandrene (9.81%) and 1,8-cineole (7.85%). The chemical composition of the essential oil reflects what is generally reported by related literature [16], hence with sesquiterpenes oxygenated, typical of the genus, as the most representative chemical constituents and ar-turmerone as the leading compound, followed by α- and β-. In general, monoterpene hydrocarbons and oxygenated ones are instead quantitatively lower. All the other compounds were detected with values lower than 3.0%, or in traces. The *Z. officinale* EO showed a similar pattern to that of *C. longa* as far as monoterpenes and sesquiterpenes are concerned, but it exhibited a prevalence of hydrocarbon compounds for both chemical classes. A-zingiberene (15.45%) and camphene (14.72%) were the most abundant compounds among sesquiterpenes and monoterpenes, respectively. The chemical composition of *Z. officinale* EO from Amazonian Ecuador reflects interesting and important differences from analogous EOs from *Z. officinale* plants of different geographical origins, reporting ar-curcumene or citral or α- and β-zingiberene as the major constituents [17]. The emerging qualitative and quantitative differences stress the particularity of the Amazonian area where the plants were grown, whose environmental characteristics significantly affect the phytochemistry of the specie. This aspect is further emphasised by the chemical characteristics of *O. campechianum* EO, which showed a eugenol chemotype with a composition quantitatively different from the same EOs from wild plants collected in southeastern Ecuador (Morona-Santiago Region) and in the Pastaza Region of Ecuador studied in the past from our research group [18,19,20]. In fact, the EO of *O. campechianum* considered in this study showed a eugenol abundance of 50.97% followed by E-caryophyllene at 10.21% and 1,8-cineole at 7.36%. Other major compounds detected were β-elemene (4.85%), bicyclogermacrene (4.11%), and spathulenol (4.42%). The most abundant compounds in *C. citratus* were geraniol (39.43%) and citral (14.37% neral and 17.29% geranial); geranyl acetate (7.96%), citronellal (4.54%), and apiole (5.02%) were less abundant; all the other compounds were lower than 3.0%, including nerol (2.64%). This pattern is partially in contrast with those reported in literature where it is indicated a higher amount of citral (approximately 60–75%) and myrcene (approximately 4–10%), not detected in our EO sample [21,22,23].

The EOs were also characterised for their vapour phase fractions, also called Headspace (HS), through HS-GC-MS (Table 3). *C. longa* and *Z. officinale* vapour phases were mainly composed of monoterpene hydrocarbons and oxygenated, 98.97% and 98.74%, respectively, with an expected lower amount of sesquiterpenes due to their lower volatility (1.03% and 1.26%, respectively). In particular, the *C. longa* EO vapour phase was characterised by limonene and 1,8-cineole (37.35%) and α-phellandrene (32.33%) with a lower abundance of sabinene (9.76%), α-pinene (8.28%), and terpinolene (7.13%). *Z. officinale* showed mainly camphene (50.39%), limonene and 1,8-cineole (17.20%), and α-pinene (16.47%) with minor amounts of β-pinene (4.40%) and myrcene (4.36%). *C. citratus* headspace was found to contain only monoterpenes (91.44%), the majority of which were oxygenated compounds (61.52%) followed by hydrocarbon ones (29.92%). *Cis*-isocitral (15.27%), γ-terpineol (12.92%), citronellal (11.07%), and trans-isocitral (10.89%) were the most abundant monoterpenes while, in decreasing order of abundance, methyl-5-hepten-one (7.11%), *trans*-ocimene (6.64%), limonene (5.30%), *cis*-ocimene (4.76%), camphene (3.49%), neral (3.45%), and nerol (3.15%) were detected. Among all the EOs examined, the headspace composition of *O. campechianum* exhibited a higher amount of sesquiterpenes (10.08%), mainly characterised by E-caryophyllene (4.95%). However, the monoterpenes were also the most characterising fraction of the *O. campechianum* EO vapour phase (89.94%), mainly characterised by 1,8-cineole and limonene (30.83%), *cis*-ocimene (19.49%), β-pinene (10.61%), *allo*-ocimene (8.43%), eugenol (7.01%), and α-pinene (4.66%). To the best of our knowledge, this is the first report about the headspace composition of the EOs of Amazonian *C. longa*, *C. citratus*, *O. campechianum*, and *Z. officinale*. In fact, among the EOs studied, only some turmeric species, including *C. longa*, from different Asian geographical areas, have been described for the composition of the volatile fractions of their EOs. The EOs of *C. longa* of different Asian geographical origins evidenced a headspace composition very different in quality and amounts from that checked in our turmeric headspace samples and characterised by compounds with large concentration ranges in relation to the different harvesting areas. In particular, the most abundant compounds were turmerone (range: 11.6–41.3%) zingiberene (range: 3.4–20.7%), β-turmerone (range: 4.1–18.0%), β-sesquiphellandrene (range: 2.3–16.1%), ar-turmerone (range: 3.2–12.4%), germacrene (range: not detected-5.3%), β-bisabolene (range: not detected–3.0%) [24]. Therefore, these results represent the first report on the headspace composition of Amazonian *C. longa*, *C. citratus*, *O. campechianum*, and *Z. officinale* EOs, highlighting once more how these phytocomplexes are particularly sensitive to geographical and environmental conditions, significantly varying their composition and, consequently, their bioactive properties.

### 2.2. Antioxidant Activity of Essential Oils: DPPH and ABTS Assays

The assessment of the biological activity of the various EOs considered the evaluation of the radical-scavenger antioxidant activity by means of spectrophotometric DPPH (1,1-diphenyl-2-picrylhydrazyl) and ABTS (2,2′-azino-bis-3-ethylbenzothiazoline-6-sulphonic acid) assays. The results were compared with those obtained from the commercial EO of *Thymus vulgaris*, a reference phytocomplex, and with that of Trolox^®^, a proven pure antioxidant compound (Table 4). The only EO that showed interesting antioxidant activity with both tests was that of *O. campechianum*. In particular, this EO evidenced a slightly higher IC_50_ in both DPPH and ABTS assays than the Trolox^®^. However, the most relevant result regards the comparison of the antioxidant capacity of *O. campechianum* EO with that of the commercial *Thymus vulgaris* EO. In fact, the IC_50_ was about 27 times better than that of thyme with the DPPH test, and more than 220 times better with the ABTS assay. These results confirmed previous evidence that emerged through several antioxidant tests performed on *O. campechianum* EO obtained from wild plants harvested in the same Amazonian area, pointing out that the cultivated plants reflect the same chemical and biological properties, suggesting their important applicative perspective as a source of phytocomplexes with radical-scavenging properties [18]. *Z. officinale* and *C. longa* EOs showed interesting antioxidant capacities only with the ABTS test but, in any case, their bioactivity was lower than that of the *T. vulgaris* OE used as the positive control. In literature, *Z. officinale* EO from Ecuador demonstrates interesting antioxidant capacities with different in vitro assays; thus, our results are not significantly comparable, suggesting only an interesting activity against the ABTS radical [25]. 

The EO of Indian *C. longa* has been studied for its in vitro antioxidant properties advising its potential health benefits as a radical scavenger, but also for its significant anti-inflammatory and antinociceptive activities [26]. Similarly, a publication on the EO from Brazilian *C. longa* cultivated plants underlines a good antioxidant activity emerged with the same methods used for our samples using BHT (Butylated hydroxytoluene) as a positive control but detecting radical scavenging values similar to ours [27]. However, concerning our results and the positive controls, especially thyme EO, we cannot come to the same conclusions for our Ecuadorian samples. As regards the *C. citratus* EO, with specific reference to south American samples, the literature suggests interesting uses as a natural additive in the food industry, mainly for flavouring, against significant antioxidant results that we have not registered [28]. Therefore, given the scarce efficacy that emerged from our results, only its use as a food flavouring can be hypothesised but, for our samples, no other functional properties can be suggested, except for the EO of *O. campechianum*.

Considering the significant results regarding the EO of *O. campechianum*, HP-TLC bioautography was performed to detect the compound(s) mainly responsible for the antioxidant capacity against DPPH and ABTS radicals (Figure 1). In plates prepared and eluted as reported in [12] treated with the free radicals DPPH (Figure 1c) and ABTS (Figure 1d) at R_f_ 0.5 an evident and a wide bleached band appeared due to the antioxidant capacity of eugenol. In fact, at R_f_ 0.5 appeared a wide yellow band (Figure 1a) corresponding to eugenol, as demonstrated by the elution of the pure standard and related literature [29]. The result accounts for the important and documented antioxidant properties of eugenol, detected as the most abundant compound in our *O. campechianum* EO (Table 2), prodromal for numerous therapeutic properties with respect to mechanisms of action and at a pre-clinical level [30].

### 2.3. Antimicrobial Activity: MIC of the EOs and Growth Inhibition Percentage of Their Headspace Fractions

The antimicrobial activities of the EOs from studied species were verified both employing the disk-diffusion assay and the agar vapour method (Headspace’s bioactivity) to check the efficacy of the whole essential oil and that of the most volatile fraction, respectively (Table 5 and Figure 2A–D). The bioactivity was checked against Gram-positive and Gram-negative bacteria, some non-pathogenic but potential food contaminants, others generally considered as nosocomial pathogens, especially in immunocompromised patients [31], and against the yeasts *Candida albicans* and *Saccharomyces cerevisiae*. 

In general, the MIC values of all the EOs were always higher than those shown by *T. vulgaris* (positive control), except for the EO of *O. campechianum* against the Gram-negative bacteria *K. oxytoca* (MIC = 0.75 mg/mL), *E. coli* (MIC = 1.70 mg/mL) and *P. aeruginosa* (MIC = 1.70 mg/mL). These results are particularly relevant because they highlight the possible use of *O. campechianum* EO as a new tool both to counteract infections caused by widespread and important gram-negative bacteria, generally considered more resistant to antibiotic treatments than gram-positive ones, and to meet the pressing need to contrast the increasingly emerging antibiotic resistance [32]. Moreover, the bacterial strains more sensitive to *O. campechianum* EO can cause intestinal (e.g., abdominal pain, vomiting, bloody diarrhoea) and extra-intestinal (e.g., urinary tract infections, peritonitis, septicaemia, pneumonia, and meningitis) diseases, up to sepsis in the worst cases. These possible consequences stress once more the importance of the antimicrobial results of *O. campechianum* EO [33,34,35]. Given the promising results, an HP-TLC antimicrobial bioautographic assay was performed on *K. oxytoca* to detect the main effective compound(s) (Figure 3). The R_f_ 0.5 band showed a clear zone of growth inhibition, highlighting that eugenol is largely responsible for the antibacterial activity, in line with what has been widely reported in the literature [36]. Additionally, *C. citratus* EO displayed an interesting antimicrobial activity, in particular against the gram-negative bacteria *E. coli* and *P. aeruginosa* in which the MIC was however comparable with what was detectable for the EO of *Thymus vulgaris* (positive control). However, the bioactivity results were consistently lower than those shown by *O. campechianum*, and partially in contrast with what was reported in the literature, where the antimicrobial activity of *C. citratus* (EO and extracts) was found to be noteworthy against different bacterial strains, suggesting promising application developments [37]. *C. longa* EO did not show any interesting antimicrobial activity evidencing the worst MIC values detected among all the EOs, in line with what was reported in the literature, in which the commercial EO displayed biological activity only in synergistic association with ascorbic acid [38]. Finally, the studied *Z. officinale* EO showed very weak bioactivity, resembling that of *C. longa* EO but in contrast to what is reported in the literature, where it is described as a phytocomplex that provides a broad antimicrobial spectrum against different microorganisms, making it an interesting alternative to synthetic antimicrobials [39].

The vapour phase of the EOs from the studied species was also checked to verify possible differences in the antimicrobial activities of the more volatile fractions (Figure 2). The results reflected what is evidenced by Table 5, indicating *O. campechianum* as the most effective, at all doses tested, of the EOs studied, particularly against the Gram-negative bacteria *K. oxytoca*, *E. coli*, and *P. aeruginosa*. At the dose of 5μL, the total growth inhibition of *K. oxytoca* stands out, as well as the 80% growth inhibition of *E. coli* and *P. aeruginosa* by the OE of *O. campechianum* compared to the bioactivity of *T. vulgaris* towards the same strains, whose growth was inhibited by 80%, 40%, and 30%, respectively. The bioactivity of the other EOs was significantly lower, with the EO of *C. citratus* being less active than that of *O. campechianum*, followed by that of *Z. officinale* and *C. longa*.

### 2.4. Cytotoxicity and Mutagen Protection Properties of the Amazonian EOs

To investigate the possible mutagen-protective capacity of the EOs from studied species, a properly modified Ames test was performed as reported in [12,13] using *Salmonella typhimurium* tester strains TA98 and TA100, with and without metabolic activation (S9 mix). 2-nitrofluorene (2 μg/plate, NF) and sodium azide (1 μg/plate, SA) were employed as mutagen inducers for TA98 and TA100 without metabolic activation, while 2-aminoanthracene (2 μg/plate, AA) was used for the same strains in presence of S9 mix. To avoid overlapping cytotoxic and antimutagenic effects, pointing out the evidence that the mutant colonies are not a result of cell-killing, a survival assay was performed for the treated bacteria to evaluate the Highest Uneffective Dose (HUD), i.e., the maximum concentration of the EOs that do not cause cytotoxicity [40] (Table 6A–D). All the EOs from the studied species evidenced significant cytotoxicity at concentrations higher than 1.00 mg/plate. This result led us to evaluate the mutagenic protection at a concentrations range of EOs between 0.00 and 1.00 mg/plate (Table 7A–D). 

*Z. officinale* EO showed a mutagen-protective capacity starting from the concentration of 0.20 mg/plate for TA98 strain against 2-nitrofluorene (NF) without metabolic activation. For the TA98 strain against 2-aminoantracene (AA) with metabolic activation, and TA100 strain against 2-aminoantracene and sodium azide (SA), with and without S9 respectively, the mutagen protection was instead displayed starting from the highest concentrations 0.50–1.00 mg/plate) (Table 7A). In the literature, there is no evidence of antigenotoxic properties of the EO of *Z. officinale*, but there are instead results about mutagen–protective properties of other kinds of extracts—e.g., aqueous ones—and their role in preventing the onset of metabolism-related pathologies such that diet-induced in pre-clinical tests [41].

*C. longa* EO, known to be no-genotoxic [42], instead evidenced a significant mutagen-protective capacity from 0.10 to 1.00 mg plate with TA98 against NF and TA100 against AA, without and with metabolic activation, respectively (Table 7B). This result is partly consistent with what is reported in the literature, where it is mainly reported about curcumin-rich aqueous extracts and derivatives and their protective role in nutrition, and their possible contribution to reducing the onset of tumours [43].

*O. campechianum* EO, already checked for its genotoxic potential through Ames tests in our previous research [20], showed a significant and very interesting mutagen-protective capacity starting from the concentration of 0.02 mg/plate for TA98 strain with and without metabolic activation. Lower but significant mutagen-protective capacity was shown for the TA100 strain starting from the concentration of 0.1 mg/plate and 0.2 mg/plate in S9 mix activated and non-activated cultures, respectively (Table 7C). 

*C. citratus* EO is reported in the literature to have a genotoxic protective effect in pre-clinical studies [44] and our results confirmed the assumption evidencing mutagen protective effect since 0.10 mg/plate for both TA98 and TA100 strains with metabolic activation (Table 7D).

To the best of our knowledge, this is the first report about the mutagen-protection capacity of the EOs from *Z. officinale*, *C. longa*, *O. campechianum*, and *C. citratus* grown in the Amazonian area. Among the four EOs, O. campechianum in particular showed the best in vitro results with the Ames’ *Salmonella* strains opening important perspectives for its use, e.g., as a food additive for its protective properties against potential mutagens. 

## 3. Materials and Methods

### 3.1. Plant Material and Isolation of Essential Oil

Fresh plant parts from three different sampling of *Zingiber officinale* (Zingiberaceae; rhizome), *Curcuma longa* (Zingiberaceae; rhizome), *Ocimum campechianum* (Lamiaceae; aerial parts), and *Cymbopogon citratus* (Poaceae; aerial parts), were collected in the same field at the balsamic period from plants grown in an Amazonian Ecuador area cultivated by the Fundaciòn Chankuap (Macas, Morona-Santiago province, Ecuador). Fresh plant material of each specie (approximately 7 kg) was processed by a 3 h steam distillation using a Clevenger’s type mobile essential oil distiller (Essential Oil Company, Portland, OR, USA). Three different distillations were performed for each plant species. The essential oils (EOs) were then dried over anhydrous sodium sulphate and stored in airtight glass vials with teflon-sealed caps at −18.0 ± 0.5 °C in the dark to prevent degradation before analyses [12,45].

### 3.2. Chemicals

All the solvents employed for chemical analyses and bioassays were chromatographic grades. Solvents and pure compounds were all purchased from Sigma–Aldrich Italy (Milano, Italy). All the microbial culture media were from Oxoid Italia (Garbagnate, Italy). Commercial essential oil of *Thymus vulgaris* (limonene chemotype) was purchased from Extrasynthese (Genay, France) and employed as an experimental positive reference for bioassays [21,46]. Lyophilised post-mitochondrial supernatant S9 fraction (Aroclor 1254-induced, Sprague–Dawley male rat liver in 0.154 MKCl solution), commonly used for the activation of pro-mutagens to mutagenic metabolites, was purchased from Molecular Toxicology, Inc. (Boone, NC, USA) and stored at −80 °C. The components of the S9 mix were: 8 mM MgCl_2_, 32.5 mM KCl, 5 mM G6P, 4 mM NADP, 0.1 m sodium phosphate buffer pH 7.4, and S9 at the concentration of 0.68 mg/mL of the mix.

### 3.3. Gas Chromatography—Flame Ionization Detector (FID)

EO samples were analysed and the relative peak areas for individual constituents were averaged. The relative percentages were determined using a ThermoQuest GC-Trace gas-chromatograph equipped with a FID detector and a Varian FactorFour VF-5ms poly-5% phenyl-95%-dimethyl-siloxane bonded phase column (i.d., 0.25 mm; length, 30 m; film thickness, 0.25 µm). Operating conditions were as follows: injector temperature 280 °C; FID temperature 300 °C, carrier gas (Helium) flow rate 1 mL/min and split ratio 1:50. Oven temperature was initially 55 °C and then raised to 90 °C at a rate of 1 °C/min, then raised to 250 °C at a rate of 10 °C/min and finally held at that temperature for 15 min. Each sample was dissolved in CH_2_Cl_2_ (1:100 *v*/*v*) and 1 μL was injected. The percentage composition of the EOs was computed by the normalisation method from the GC peak areas, calculated by means of three injections from each EO, without using correction factors.

### 3.4. Gas Chromatography—Mass Spectrometry

The analyses of the volatile compounds were performed on a GC-3800 Varian gas chromatograph coupled to a Varian MS-4000 mass spectrometer using electron impact and hooked to NIST and ABREG libraries. The constituents of the volatile oils were identified by comparing their GC retention times, LRI and the MS fragmentation pattern with those of other EOs of known composition, with pure compounds and by matching the MS fragmentation patterns and retention indices with the above-mentioned mass spectra libraries and with literature data [47]. The GC conditions were the same as reported for GC analysis (Section 3.3) and the same column was used. The MS conditions were instead as follows: ionisation voltage, 70 eV; emission current, 10 µAmp; scan rate, 1 scan/s; mass range, 40–400 Da; trap temperature, 150 °C, transfer line temperature, 300 °C. A mixture of aliphatic hydrocarbons (C8–C24) in hexane was injected under the above temperature program to calculate the arithmetic indices [12].

### 3.5. Headspace Gas Chromatography—Mass Spectrometry

The chemical composition of the volatile fraction of EOs was determined by static headspace analysis in GC-MS under the same conditions abovementioned (Section 3.3 and Section 3.4). Five hundred microliters of each sample were placed in an 8 mL vial sealed with a crimp top and kept at 37.0 ± 0.5 °C for 1 h. The vapour phase was drawn off with a 1 mL gas-tight syringe and injected into the gas chromatograph [46].

### 3.6. Biological Activities of Essential Oils (EOs)

Antioxidant, antimicrobial, and antimutagenic capacities were evaluated for all the EOs. All the bioactivities were performed by comparing all the data with those achieved with commercial *T. vulgaris* essential oil considered as a positive control [46,48]. Data reported for each assay are the average of three determinations of independent experiments. For the most interesting results regarding antioxidant and antimicrobial activities, the DPPH and ABTS High-Performance Thin Layer Chromatography (DPPH/ABTS-HP-TLC) bioautographic assays were performed to point out the most effective compounds and/or compound categories [12].

#### 3.6.1. Antioxidant Properties

Radical scavenging properties were performed through 1,1-diphenyl-2-picrylhydrazyl (DPPH) assay; 2,2′-azino-bis-3-ethylbenzothiazoline-6-sulphonic acid (ABTS) spectrophotometric assay. Each EO (0.5 mL/mL in Tween 40, 0.5% *w*/*w* in distilled water) was diluted 2, 5, 10, 50, 100, and 200-fold with DMSO. The antioxidant capacity of each EO was evaluated by its ability to bleach the 1,1-diphenyl-2-picrylhydrazyl (DPPH) and 2,2′-azinobis-3-ethylbenzothiazoline-6-sulfonic acid radicals (ABTS) [49]. An aliquot of 100 µL of each solution was added to 2.9 mL of 1,1-diphenyl-2-picrylhydrazyl (DPPH; 1 × 10^−4^ M in ethanol), shaken vigorously and kept in the dark for 30 min at room temperature. Sample absorbance was measured at 517 nm with UV–vis spectrophotometer (Thermo-Spectronic Helios γ, Cambridge, UK).

A stock solution of ABTS^•+^ was produced by the reaction of ABTS solution (2 mM) with 70 mM potassium persulfate (both solutions were prepared in double-distilled water) for 12–16 h, in the dark at room temperature. The stock solution was diluted in Phosphate Buffered Saline (PBS) to achieve an absorbance of 0.70 ± 0.02 at 734 nm with a UV/VIS spectrophotometer (Thermo-Spectronic Helios γ, Cambridge, UK). An aliquot of 900 µL of diluted ABTS solution was mixed with 100 µL of the sample. The absorbance at 734 nm was taken exactly 1 min after initial mixing.

For both assays, a negative control was assessed as the solution assay described above without the EOs, instead of which Tween 40 was employed. Trolox^®^ and commercial *T. vulgaris* essential oil, known for their antioxidant capacity as pure compound and phytocomplex, respectively, were used as positive controls and prepared as described above for the studied EOs.

All the radical scavenging activities were reported as IC_50_ for each sample [46,49].

#### 3.6.2. Antimicrobial Activity

The antimicrobial activity was checked against Gram-positive and Gram-negative bacteria (*Staphylococcus aureus* subsp. *aureus* ATCC 29213, *Enterococcus faecalis* ATCC 29212, *Micrococcus luteus* ATCC 9622, *Listeria grayi* ATCC 19120, *Pseudomonas aeruginosa* ATCC 17934, *Klebsiella oxytoca* ATCC 29516, *Escherichia coli* ATCC 4350, *Proteus vulgaris* ATCC 6361), and against two yeasts (*Candida albicans* ATCC 48274 and *Saccharomyces cerevisiae* ATCC 2365) following the procedures reported in [11,20,30]. Chloramphenicol (CPh; Sigma-Aldrich) and fluconazole (Flu; Sigma-Aldrich; St. Louis, Missouri, USA) were used to test the sensitivity of bacterial and yeast strains, respectively. The EOs bioactivity was assayed employing the standard disk diffusion method: mother cultures of each micro-organism were set up 24 h before the assays to reach the stationary phase of growth. The tests were assessed by inoculating Petri dishes from the mother cultures with proper sterile media, to obtain a concentration of 10^5^ CFU/mL for bacteria and 10^6^ CFU/mL colony for yeasts. An aliquot of dimethyl sulfoxide (DMSO) was added to the EOs to obtain a 0.01–100 mg/mL concentration range. Serial dilutions of the DMSO/EO solution were deposited on sterile paper discs (6 mm diameter, Difco) which were then placed in the centre of the inoculated Petri dishes. Therefore, the Petri dishes were then incubated at 37 °C for 48 h for bacteria and 72 h for yeasts, and the growth inhibition zone diameter (IZD) was measured to the nearest mm. The lowest concentration of each DMSO/EO solution deposited on the sterile paper disc showing a clear zone of inhibition was taken as the minimum inhibitory concentration (MIC) [31,49].

Moreover, the biological activity of each EOs against the above-mentioned bacterial and yeast strains was performed by means of the properly modified agar vapour method to check the bioactivity of the most volatile fraction [46]. The microorganisms were grown in Petri plates (90 mm) supplemented with 15 mL/plate of appropriate agarised medium (5 × 10^2^ CFU/mL for bacteria and yeasts; approx. 500 CFU/plate). Sterilised filter paper disks (diameter 9.0 mm) were absorbed with pure 1–10 μL of each EO and placed inside the upper lid of each plate, at about 4 mm from the strain. Plates were kept in an inverted position, tightly sealed with parafilm, and incubated for 48 h for bacteria and 72 h for yeasts at 37.0 ± 1.0 °C. Blanks served as a negative control. Three replicates were made for each treatment and the results were expressed as growth inhibition percentage based on the number of colonies counted for each plate with reference to negative control where the growth was considered as 100%. The strains were cultured in nutrient agar, tryptic soy agar, and YEPD (yeast extract, peptone, and dextrose) following the suggestions given by ATCC protocols (www.ATTC.org; accessed in 1 June 2021).

#### 3.6.3. HP-TLC-Bioautographic Assay of the Antioxidant and Antimicrobial Most Interesting Results

To point out the main EO compounds as separate bands on TLC plates, ten microliters on a 6 mm band of 1% solutions of each EO in hexane were deposited to HP-TLC (plates silica gel 60 F_254_, Merck) precoated plates (90 × 10 mm), developed in a chromatographic chamber with toluene/ethyl acetate/petroleum ether 93/7/20, and dried for complete removal of solvents. Detection was achieved by UV light (254 nm) and then monitored at visible light after spraying in sequence with solutions of 10% H_2_SO_4_ ethanolic and 2% vanillin ethanolic, followed by heating at 105 °C for 10 min [29].

To perform High-Performance Thin Layer Chromatography (HP-TLC) bioautographic assay of the most relevant results emerged by radical scavenging activity spectrophotometric assays using the DPPH and the ABTS radicals, 10 μL of each EO hexane solution (1% *v*/*v*) was applied in triplicate to a 90 × 100 mm HP-TLC plate of silica gel (plates silica gel 60 F_254_, Merck) as 10 mm wide bands with Linomat V (Camag). Then, the same spots deposited on two distinct plates were eluted as above described. After elution, one plate was sprayed with an ethanolic solution of DPPH (2 mg/mL), while the other was sprayed with ABTS stock solution prepared as above-described (Section 3.6.1) to detect the antioxidant fractions against the two different radicals. Antiradical compounds appeared as clear white spots against a violet-coloured background for the HP-TLC-DPPH plate, while the background appeared green-coloured for the (HP)TLC-ABTS plate [12].

To perform HP-TLC bioautographic assay to point out the bioactive antimicrobial compounds and/or compound categories that emerged in the most interesting results, HP-TLC plates were prepared as described in Section 3.6.2 and treated as reported in [12]. After 24 h incubation, the results led to point out the effective fractions (bands) of the EOs, which determined the presence of localised microbial growth inhibition zones on the eluted HP-TLC plates.

Related literature consultation [29], the elution of pure compounds putative responsible for bioactivity on HP-TLC plates, and the analyses by GC-MS of the scratched-out and ethyl acetate extracted bioactive bands were the strategies adopted to identify the bioactive spots corresponding to active constituents.

#### 3.6.4. Amazonian Essential Oils Cytotoxicity and Mutagen Protection Properties: Highest Uneffective Dose (HUD) and Ames Test Properly Modified

The inhibitory effect of the Amazonian EOs samples (concentration range 0.0–1.0 mg/plate) on the mutagenic activity of direct-acting mutagen 2-nitrofluorene (2 μg/plate) and sodium azide (1 μg/plate), was examined in plate incorporation assay, derived from mutagenicity test as described by [40] with some minor modifications, using tester strain TA98 and TA100, respectively. The inhibitory effect of the EOs on the mutagenic activity of the indirectly acting mutagen 2-aminoanthracene (2 μg/plate) was instead examined in a plate incorporation assay, using tester strain TA98 and TA100 with S9 mix. The inhibition rate for mutagenic induction was calculated according to the formula:$$Inhibition\;rate\;\left(\%\right)=\frac{\left(A-B\right)}{A}\times 100$$
where *A* represents the revertants in the positive control and *B* the revertants in the EOs samples, having subtracted the spontaneous revertants. A critical point, affecting the outcome of the interaction between an antimutagen and a testing bacterial strain is the overlapping of the cytotoxic and antimutagenic dose concentrations. In other words, it is important to confirm that the dose-dependent disappearance of the mutant colonies is not a result of cell killing. For this purpose, a simple survival assay for the treated bacteria must be performed to evaluate a Highest Uneffective Dose (HUD). To verify the toxicity of the analysed samples on bacterial cells and evaluate the HUD, a toxicity test was performed [40]. A fresh 15 h culture was diluted to give a 1–2 × 10^4^ bacteria/mL. The test samples at several concentrations (10^−2^, 5 × 10^−2^, 10^−1^, 5 × 10^−1^, 1, 5, 10 mg/plate) diluted in DMSO, mixed with 2 mL of molten top agar, were plated with 0.1 mL of the diluted culture. Histidine/biotin agar plates were enriched with 10 μmoles of L-histidine and 0.05 μmoles of biotin by incorporating these nutrients into the soft agar overlay. Triplicate plates were poured for each dose of solution. The colony-forming units (CFU) were assessed after the plates were incubated at 37 °C for 48 h and compared with that of the control, where no test samples were added. HUD for the EOs, with and without metabolic activation, was evaluated by visual estimation (colonies counting) integrated by statistical analyses.

### 3.7. Statistical Analysis

Relative standard deviations and statistical significance were evaluated using Student’s *t*-test (*p* < 0.05). The statistical evidence performed on the Ames test was combined with the Highest Uneffective Dose (HUD); the results were then used to interpret the results of the significant decrease in the number of *Salmonella* revertants. When the modulator dose concentration was statistically effective and it ranged below or coincided with the HUD, the samples were considered to present a sign of the effect (antimutagenicity). All computations were carried out using the statistical software STATISTICA 6.0 (StatSoft Italia Srl, Vigonza, Padova, Italy).

## 4. Conclusions

The present study reported the chemical characterisation of the essential oils (EOs) of *Zingiber officinale* (Zingiberaceae; rhizome), *Curcuma longa* (Zingiberaceae; rhizome), *Ocimum campechianum* (Lamiaceae; aerial parts) and *Cymbopogon citratus* (Poaceae; aerial parts) obtained from plants cultivated in the southeastern area of Ecuador, evidencing interesting differences in comparison with analogous literature reports—regarding both EOs from wild and cultivated plants—pointing out the peculiarity of the Ecuadorian Amazonian area in affecting the EO composition of aromatic plants. The chemical characterisation was also extended to the vapour phase fraction showing the prevalence of monoterpenes in all four EOs, but with a significant further presence of sesquiterpene hydrocarbons in the EO of *O. campechianum.*

The antioxidant activity of the EOs revealed *O. campechianum* phytocomplex as the most interesting, showing with both DPPH and ABTS assays IC_50_ values better than those shown by the *T. vulgaris* EO taken as positive reference, and comparable to those of the positive control characterised by the single compound Trolox^(R)^. The HP-TLC bioautographic assay pointed out eugenol as the compound responsible for radical scavenging bioactivity. Assessments of antimicrobial activity performed on both the EOs and their volatile fraction showed MIC values above or comparable to those of the positive control *T. vulgaris* EO. *O. campechianum* EO showed the most interesting MIC results, particularly against the Gram-positive *L. grayi* and, against the Gram-negative *K. oxytoca*, *P. aeruginosa*, and *E. coli*, suggesting interesting developments in the field of natural antimicrobials to counter the emerging and increasingly serious problems of antibiotic resistance. The bioactivity of the other EOs was significantly lower, with the EO of *C. citratus* being less active than that of *O. campechianum*, followed by that of *Z. officinale* and *C. longa*. The same results were confirmed by the antimicrobial activity evaluated on the vapour phase fractions of the EOs. As for the antioxidant assays, the bioautographic HP-TLC assay of the most performing EO, *O. campechianum*, confirmed the essential role of eugenol as the bioactive compound. Regarding the assessment of the mutagenic protective capacity of the EOs evaluated with the Ames test (TA98 and TA100 *Salmonella typhimurium* strains towards the mutagens nitrofluorene, aminoanthracene, and sodium azide), the *O. campechianum* EO highlighted the interesting capacity of inhibiting the development of mutant colonies starting from a concentration of 0.02 mg/plate. The EOs of *C. citratus*, *C. longa*, and *Z. officinale*, even if with a proportionally lower efficacy (protective capacity starting from the concentration of 0.10–0.20 mg/plate), were found to be noteworthy. Finally, to the best of our knowledge, this is the first report about the mutagen-protection capacity of the Amazonian *Z. officinale*, *C. longa*, *O. campechianum*, and *C. citratus* and the results, even if with evident differences in efficacy, suggest important perspectives for their use, e.g., as food additives for the protective properties against potential mutagens.

## Figures and Tables

**Figure 1 antibiotics-12-00177-f001:**
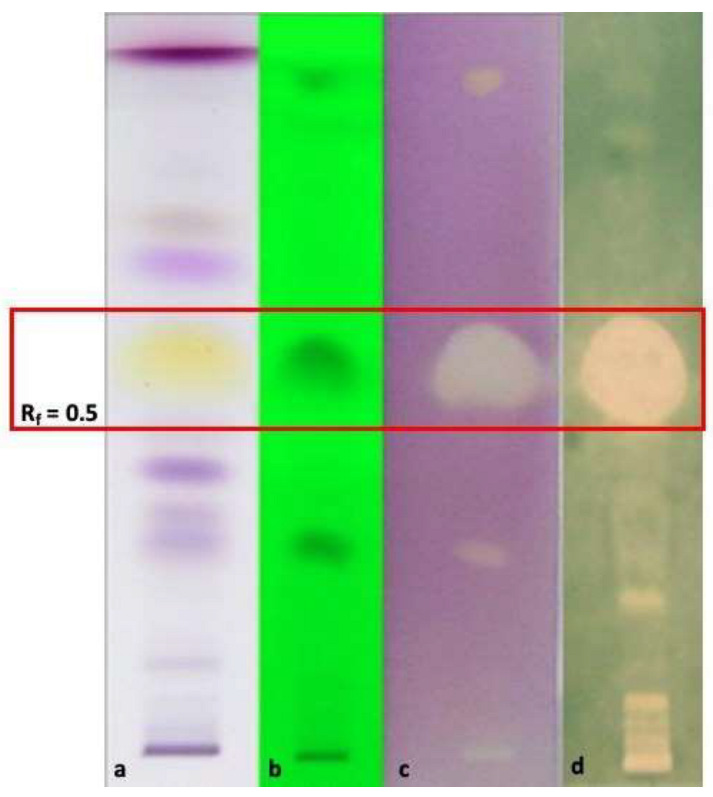
HP-TLC bioautographic assay to check the antioxidant capacity of *O. campechianum* EO against DPPH and ABTS radicals. (**a**): plate eluted and treated with vanillin sulphuric acid reagent; (**b**): plate eluted and visualised at λ = 254 nm wavelength; (**c**): plate eluted ad treated with DPPH; (**d**): plate eluted and treated with ABTS. The band at R_f_ = 0.5 corresponds to eugenol, as verified eluting pure standard compound (data not shown) and comparing the result with related literature [29]. The large bleached bands at R_f_ = 0.5 correspond to the capacity of eugenol to contrast the oxidative reactivity of the DPPH• (3) and ABTS•+ (4) radicals.

**Figure 2 antibiotics-12-00177-f002:**
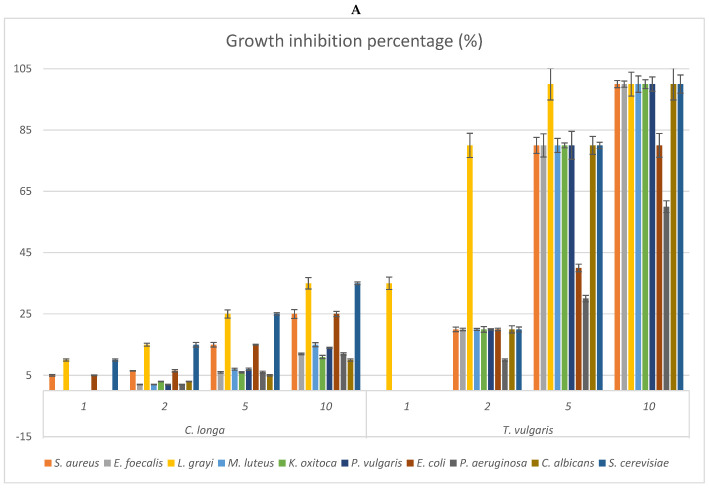
(**A**–**D**) Antimicrobial activities of the vapour phase of the EOs from studied species compared to that of commercial *T. vulgaris*. The bioactivity is expressed as growth inhibition percentage (%) checked at the doses of 1, 2, 5, and 10 μL. The bioactivity of the vapour phase of each EO is compared to that of *T. vulgaris* (positive control). (**A**) Growth inhibition percentage of the vapour phase of *Curcuma longa*; (**B**) growth inhibition percentage of the vapour phase of *Cymbopogon citratus*; (**C**) growth inhibition percentage of the vapour phase of *Ocimum campechianum*; (**D**) growth inhibition percentage of the vapour phase of *Zingiber officinale.* Values are averages ± standard deviation.

**Figure 3 antibiotics-12-00177-f003:**
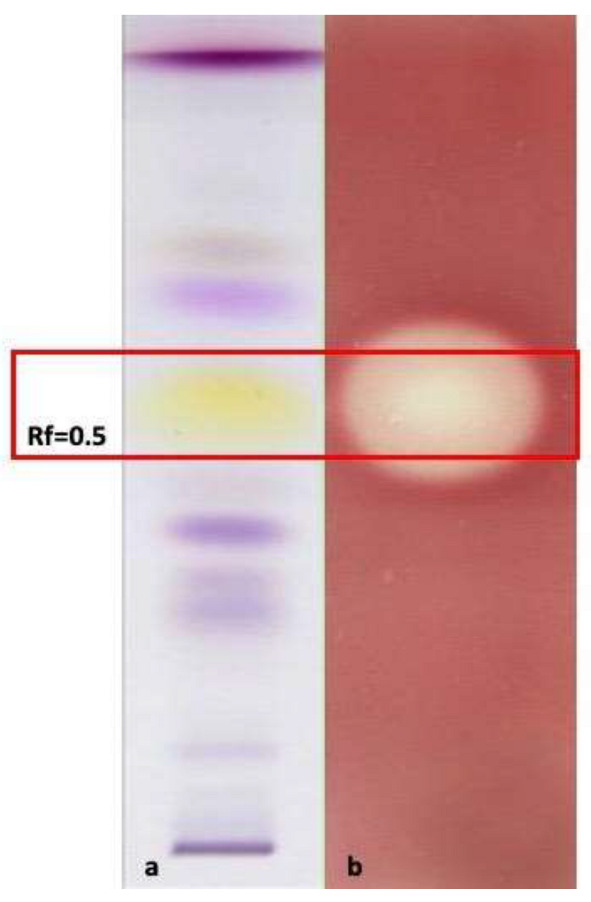
HP-TLC bioautographic assay showing the antimicrobial activity of *O. campechianum* EO, the most interesting among the EOs tested. The bioautographic plate (**b**) shows the bleaching band (microbial growth inhibition zone) of the antimicrobial activity against *Klebsiella oxytoca*, the most sensitive bacterial strain with the lowest MIC detected (0.75 mg/mL), at Rf = 0.5, corresponding to eugenol, the compound responsible of the bioactivity for *O. campechianum* EO. (**a**) Plate eluted and treated with vanillin sulphuric acid reagent; (**b**) plate eluted and treated with *K. oxytoca* for the antimicrobial bioautographic evaluation.

**Table 1 antibiotics-12-00177-t001:** Yields and density of Amazonian essential oils (EOs) obtained by steam distillation of *Curcuma longa*, *Cymbopogon citratus*, *Ocimum campechianum*, and *Zingiber officinale* fresh plant material.

Plant Species	Part Used	Hydro-Distillation Yield (mL/Kg Fresh Plant Material)	Density (g/mL)
*Curcuma longa*	Rhizome	2.6 ± 0.4	0.8975
*Cymbopogon citratus*	Aerial parts	3.0 ± 0.5	0.8760
*Ocimum campechianum*	Aerial parts	7.0 ± 0.8	0.9310
*Zingiber officinale*	Rhizome	4.0 ± 0.6	0.8870

**Table 2 antibiotics-12-00177-t002:** Chemical composition of essential oils (EOs) from studied species through GC and GC-MS analyses: *Curcuma longa*; *Cymbopogon citratus*; *Ocimum campechianum*; *Zingiber officinale*.

N	Compound ^a^	AI ^b^	*Curcuma longa*	*Cymbopogon citratus*	*Ocimum campechianum*	*Zingiber officinale*
			Area %			
1	α-Pinene	932	0.47	- ^c^	Tr ^d^	4.16
2	Camphene	946	-	-	tr	14.72
3	Sabinene	969	1.31	-	tr	-
4	β-Pinene	974	-	-	0.49	0.47
5	Methyl-5-hepten-2-one	981	-	0.41	-	0.69
6	Myrcene	988	0.36	-	0.19	1.80
7	α-Phellandrene	1002	9.81	-	-	0.71
8	α-Terpinene	1014	0.25	-	-	-
9	p-Cymene	1020	2.17	-	-	-
10	Limonene	1024	1.61	tr	0.17	5.75
11	1,8-Cineole	1026	7.85	-	7.36	8.61
12	*cis*-Ocimene	1032	-	-	2.87	-
13	*trans*-Ocimene	1044	-	-	tr	-
14	γ-Terpinene	1054	0.46	-	-	-
15	Terpinolene	1086	2.30	-	-	0.49
16	Linalool	1095	0.26	0.46	1.87	0.95
17	*allo*-Ocimene	1128	-	-	0.16	-
18	1-Terpineol	1130	-	-	-	-
19	Citronellal	1148	0.22	4.54	-	-
20	Borneol	1165	-	-	0.19	3.59
21	*cis*-Isocitral	1160	-	0.54	-	-
22	4-Terpineol	1174	0.31	-	-	-
23	*trans*-Isocitral	1177	-	0.89	-	-
24	α-Terpineol	1186	0.47	-	0.59	1.53
25	γ-Terpineol	1199	-	-	-	-
26	Nerol	1227	-	2.64	-	-
27	Neral	1235	-	14.37	-	3.98
28	Geraniol	1249	-	39.43	-	0.44
29	Geranial	1264	tr	17.29	-	5.16
30	Bornyl acetate	1287	-	-	-	0.38
31	Thymol	1289	tr	-	-	-
32	Carvacrol	1298	tr	-	-	-
33	δ-Elemene	1335	-	-	tr	-
34	Eugenol	1356	-	-	50.97	-
35	Neryl acetate	1359	-	0.52	-	-
36	β-Elemene	1389	-	-	4.85	0.35
37	Geranyl acetate	1379	-	7.96	-	-
38	E-Caryophyllene	1417	0.38	2.46	10.21	-
39	α-Humulene	1452	-	0.41	2.05	-
40	*allo*-Aromadendrene	1458	-	-	0.53	-
41	Germacrene D	1484	-	0.47	-	1.45
42	ar-Curcumene	1479	1.22	-	-	3.80
43	β-Selinene	1489	-	-	1.66	-
44	Bicyclogermacrene	1500	-	-	4.11	-
45	*trans*-Muurola-4(14)5-diene	1493	-	-	-	2.24
46	Germacrene A	1508	-	-	2.76	-
47	α-Zingiberene	1493	1.25	-	-	15.45
48	α-Bisabolene	1506	0.27	-	-	-
49	α-(E,E)-Farnesene	1505	-	-	-	8.52
50	δ-Cadinene	1522	-	0.56	-	tr
51	β-Sesquiphellandrene	1521	2.20	-		6.87
52	Germacrene B	1559	-	-	0.79	0.95
53	*trans*-Nerolidol	1561	-		-	0.51
54	α-Cadinene	1537	-	1.13	-	-
55	ar-Turmerol	1582	1.38	-	-	-
56	Spathulenol	1577	-	-	4.42	-
57	Helifolen-12-ale A	1592	1.45	-	-	-
58	Apiole	1620	-	5.02	-	-
59	β-Biotol	1612	2.03	-	-	-
60	β-Eudesmol	1649	-	-	-	0.98
61	α-Cadinol	1654	-	0.49	-	-
62	ar-Turmerone	1668	23.35	-	-	2.06
63	α-Turmerone	1671	22.81	-	-	0.87
64	β-Turmerone	1707	15.27	-	-	0.56
Total identified			99.46	99.59	96.24	98.04
Monoterpene hydrocarbons			18.74	0.00	3.88	28.10
Monoterpene oxygenated: Alcohols:			9.11	89.05	60.98	25.33
			1.04	42.53	53.62	6.51
	Aliphatics		1.04	42.53	2.65	6.51
	Phenolics		-	-	50.97	-
	Esters		-	8.48	-	0.38
	Aldehydes and ketones		0.22	38.04	-	9.83
	Ethers		7.85	-	7.36	8.61
Sesquiterpene hydrocarbons			5.32	5.03	26.96	39.63
Sesquiterpene oxygenated			66.29	5.51	4.42	4.98

^a^ Compounds are listed in order of elution from a Varian FactorFour VF-5 ms column. ^b^ AI (arithmetic retention index) calculated on a Varian FactorFour VF-5 ms column. ^c^ tr = trace (<0.1%). ^d^ Dash indicates not detected.

**Table 3 antibiotics-12-00177-t003:** Headspace composition of Amazonian essential oils (EOs) through GC and GC-MS analyses: *Curcuma longa*; *Cymbopogon citratus*; *Ocimum campechianum*; *Zingiber officinale*.

N	Compound ^a^	AI ^b^	*Curcuma longa*	*Cymbopogon citratus*	*Ocimum campechianum*	*Zingiber officinale*
			Area %			
1	Tricyclene	921	-^d^	-	-	0.78
2	α-Pinene	932	8.28	2.24	4.66	16.47
3	Camphene	946		3.49	2.51	50.39
4	Sabinene	969	9.76	0.97	0.26	-
5	β-Pinene	974	-	-	10.61	4.40
6	Myrcene	981	1.35	-	3.24	4.34
7	Methyl-5-hepten-2-one	988	-	7.11	-	-
8	α-Phellandrene	1002	32.33	0.91	-	2.25
9	α-Terpinene	1014	-	0.70	-	-
10	Limonene	1024	8.15	5.30	1.73	8.1
11	1,8-Cineole	1026	29.2	-	29.1	9.1
12	*cis*-Ocimene	1032	-	4.76	19.49	-
13	*trans*-Ocimene	1044	0.18	6.64	0.29	-
14	γ-Terpinene	1054	1.65	2.03	-	-
15	Terpinolene	1086	7.13	0.54	0.17	0.48
16	Linalool	1095	0.16	0.78	2.11	0.25
17	*allo*-Ocimene	1128	-	2.33	8.43	-
18	Citronellal	1148	-	11.07	-	-
19	Borneol	1165	-	0.97	0.11	0.57
20	4-Terpineol	1160	0.16	1.87	-	-
21	α-Terpineol	1174	0.12	1.16	0.22	0.19
22	*cis*-Isocitral	1177	-	15.27	-	-
23	*trans*-Isocitral	1186	-	10.89	-	-
24	γ-Terpineol	1199	-	12.92	-	-
25	Nerol	1227	-	3.15	-	-
26	Neral	1235	-	3.45	-	0.57
27	Geraniol	1249	0.17	tr ^c^	-	-
28	Geranial	1264	0.14	tr ^c^	-	0.70
29	Thymol	1289	-	tr ^c^	-	-
30	δ-Elemene	1335	-	-	0.18	-
31	Eugenol	1356	0.19	-	7.01	0.15
32	β-Elemene	1389	-	-	2.88	0.07
33	E-Caryophyllene	1417	0.09	-	4.95	0.06
34	α-Humulene	1452	-	-	0.71	-
35	*allo*-Aromadendrene	1458	-	-	0.21	-
36	ar-Curcumene	1479	0.11	-	-	0.14
37	β-Selinene	1489	-	-	0.46	-
38	Bicyclogermacrene	1500	-	-	0.67	-
39	*trans*-Muurola-4(14)5-diene	1493	-	-	-	0.13
40	α-Zingiberene	1493	0.09	-	-	0.48
41	α-(E,E)-Farnesene	1505	-	-	-	0.23
42	β-Sesquiphellandrene	1521	0.08	-	-	0.15
43	ar-Turmerone	1668	0.26	-	-	-
44	α-Turmerone	1671	0.23	-	-	-
45	β-Turmerone	1707	0.18	-	-	-
Total identified			100.00	98.55	100.00	100.00
Monoterpene hydrocarbons			68.83	29.91	51.39	87.21
Monoterpene oxygenated:			30.14	68.64	38.55	11.53
	Alcohols:		0.80	20.85	9.45	1.16
	Aliphatics		0.61	20.85	2.44	1.01
	Phenolics		0.19	-	7.01	0.15
	Esters		-	-	-	-
	Aldehydes and Ketones		0.14	47.79	-	1.27
	Ethers		29.2	-	29.10	9.10
Sesquiterpene hydrocarbons			0.37	-	10.06	1.26
Sesquiterpenes oxygenated			0.66	-	-	-

^a^ Compounds are listed in order of elution from a Varian FactorFour VF-5 ms column. ^b^ AI (linear retention index) calculated on a Varian FactorFour VF-5 ms column. ^c^ tr = trace (<0.1%). ^d^ Dash indicates not detected.

**Table 4 antibiotics-12-00177-t004:** Antioxidant properties of EOs from studied species through DPPH and ABTS tests in comparison with that of commercial *Thymus vulgaris* (phytocomplex as positive control), and Trolox^®^ (pure chemical compound as positive control).

	DPPH	ABTS
Essential Oils	IC_50_ (mg/mL)	
*Curcuma longa*	16.512 ± 2.452	0.871 ± 0.132
*Cymbopogon citratus*	2.270 ± 0.340	4.322 ± 0.651
*Ocimum campechianum*	0.012 ± 0.003	0.0013 ± 0.0004
*Zingiber officinale*	5.478 ± 0.082	0.563 ± 0.080
*Thymus vulgaris*	0.325 ± 0.038	0.288 ± 0.041
Trolox^®^	0.0060 ± 0.0010	0.0024 ± 0.0003

IC_50_ = half-maximal concentration (IC_50_) to determine the antioxidant effect.

**Table 5 antibiotics-12-00177-t005:** Antimicrobial activities of EOs from the studied species compared with that of commercial *T. vulgaris*. Chloramphenicol (*CPh*) and fluconazole (*Flu*) were used to test the sensitivity of bacterial and yeast strains respectively.

		MIC (mg/mL)					
		*C. longa*	*C. citratus*	*O. campechianum*	*Z. officinale*	*T. vulgaris*	*CPh* (mg/mL)
Gram +	*S. aureus*	48.75	9.31	8.67	17.74	1.93	3.50 × 10^−3^
	*E. foecalis*	89.50	9.31	8.67	44.35	1.93	3.50 × 10^−3^
	*L. grayi*	8.98	3.50	1.70	8.87	0.97	1.80 × 10^−3^
	*M. luteus*	89.50	17.34	9.31	44.35	1.93	3.50 × 10^−3^
Gram −	*K. oxytoca*	89.75	4.34	0.75	8.87	1.93	1.80 × 10^−3^
	*P. vulgaris*	89.75	8.67	4.66	17.74	1.93	3.50 × 10^−3^
	*E. coli*	44.88	4.66	1.70	8.87	4.84	3.00 × 10^−3^
	*P. aeruginosa*	89.50	9.31	1.70	10.80	9.67	3.00 × 10^−3^
							*Flu* (μg/mL)
Yeasts	*C. albicans*	89.75	4.70	4.10	17.74	1.93	0.125
	*S. cerevisiae*	8.98	3.45	2.25	88.70	1.93	0.500

MIC: Minimum Inhibitory Concentration, considered as the lowest concentration of each DMSO/EO solution deposited on the sterile paper disc showing a clear zone of inhibition.

**Table 6 antibiotics-12-00177-t006:** (**A**–**D**). Highest Uneffective Dose (HUD) tested with and without metabolic activation (S9 mix). HUD represents the maximum concentration of EOs DMSO diluted which does not induce cytotoxic evidence in *S. typhimurium* TA98 and TA100 strains cultures. The HUD data are essential to interpret the results of a significant decrease in the number of *Salmonella* revertants in anti-genotoxic assays. Negative controls (EO = 0.000 mg/plate) were set up with 100 mL/plate of DMSO. The results are expressed both as survival percentage ± standard deviation (s.d.) and Colony Forming Units (CFU)/plate ± standard deviation. Table 6A: *Zingiber officinale* EO cytotoxicity; Table 6B: *Curcuma longa* EO cytotoxicity; Table 6C: *Ocimum campechianum* EO cytotoxicity; Table 6D: *Cymbopogon citratus* EO cytotoxicity.

A				
*Z. officinale* (mg/plate)	TA98-S9	TA98+S9	TA100-S9	TA100+S9
0.00	38.0 ± 6.0	45.0 ± 4.1	101.0 ± 11.7	147.0 ± 13.7
0.01	39.0 ± 5.1	43.0 ± 3.1	105.0 ± 7.0	141.0 ± 11.3
0.05	37.0 ± 7.4	48.0 ± 3.0	98.0 ± 5.4	133.0 ± 8.9
0.10	33.0 ± 4.9	44.0 ± 3.3	100.0 ± 5.9	144.0 ± 7.7
0.50	34.0 ± 4.7	39.0 ± 3.2	96.0 ± 5.3	138.0 ± 6.6
1.00	32.0 ± 2.4	37.0 ± 3.3	99.0 ± 6.6	140.0 ± 6.7
2.50	22 ± 2.0 *	20.0 ± 1.5 *	88.0 ± 2.7	136.0 ± 5.5
5.00	10.0 ± 0.3 *	9.0 ± 0.4 *	29.0 ± 4.1 *	119.0 ± 6.3
10.00	- *	- *	5.0 ± 0.0 *	58.0 ± 1.0 *
**B**				
** *C. longa* ** **(mg/plate)**	**TA98-S9**	**TA98+S9**	**TA100-S9**	**TA100+S9**
0.00	38.0 ± 4.1	45.0 ± 3.9	101.0 ± 6.3	147.0 ± 9.6
0.01	38.0 ± 3.9	46.0 ± 4.1	99.0 ± 9.9	145.0 ± 8.4
0.05	35.0 ± 3.1	39.0 ± 3.0	98.0 ± 8.5	143.0 ± 7.1
0.10	41.0 ± 4.3	42.0 ± 2.8	100.0 ± 4.7	144.0 ± 6.5
0.50	39.0 ± 4.7	43.0 ± 2.0	91.0 ± 6.8	141.0 ± 11.2
1.00	33.0 ± 2.0	42.0 ± 3.1	101.0 ± 5.0	133.0 ± 6.7
2.50	12.0 ± 1.0 *	40.0 ± 4.7	98.0 ± 2.7	61.0 ± 3.5 *
5.00	- *	27.0 ± 1.0 *	71.0 ± 3.1 *	33.0 ± 2.2 *
10.00	- *	7.0 ± 0.3 *	11.0 ± 0.5 *	- *
**C**				
** *O. campechianum* ** **(mg/plate)**	**TA98-S9**	**TA98+S9**	**TA100-S9**	**TA100+S9**
0.00	38.0 ± 4.1	45.0 ± 5.6	101.0 ± 9.9	147.0 ± 12.3
0.01	33.0 ± 3.7	44.0 ± 6.1	104.0 ± 12.3	138.0 ± 9.8
0.05	35.0 ± 3.5	46.0 ± 4.8	96.0 ± 10.0	140.0 ± 11.1
0.10	40.0 ± 4.0	39.0 ± 4.7	100.0 ± 8.7	143.0 ± 7.8
0.50	37.0 ± 2.8	41.0 ± 3.9	99.0 ± 9.3	152.0 ± 10.7
1.00	37.0 ± 3.4	39.0 ± 2.5	95.0 ± 6.8	140.0 ± 9.5
2.50	20.0 ± 1.1 *	35.0 ± 2.8	66.0 ± 5.1 *	93.0 ± 6.2
5.00	6.0 ± 0.3 *	11.0 ± 0.6 *	42.0 ± 3.3 *	18.0 ± 0.9 *
10.00	- *	2.0 ± 0.1 *	10.0 ± 0.7 *	- *
**D**				
** *C. citratus* ** **(mg/plate)**	**TA98-S9**	**TA98+S9**	**TA100-S9**	**TA100+S9**
0.00	38.0 ± 4.1	45.0 ± 2.7	101.0 ± 7.8	147.0 ± 13.4
0.01	39.0 ± 3.7	44.0 ± 3.8	111.0 ± 9.3	144.0 ± 12.5
0.05	37.0 ± 3.3	40.0 ± 2.9	106.0 ± 10.1	138.0 ± 10.7
0.10	36.0 ± 4.0	42.0 ± 3.2	100.0 ± 8.8	140.0 ± 11.5
0.50	39.0 ± 3.8	46.0 ± 3.8	96.0 ± 9.0	127.0 ± 13.2
1.00	35.0 ± 2.7	43.0 ± 2.8	92.0 ± 7.5	133.0 ± 8.4
2.50	33.0 ± 3.0	38.0 ± 3.0	89.0 ± 6.6	115.0 ± 9.1
5.00	30.0 ± 3.2	11.0 ± 0.6 *	53.0 ± 3.6 *	66.0 ± 4.4 *
10.00	15.0 ± 0.8 *	- *	20.0 ± 1.3 *	31.0 ± 2.6 *

(-) no Colony Forming Units (CFU) were detected because of the cytotoxicity expressed by the Amazonian essential oils (EOs). (*) Significant evidence (cytotoxicity) in light of Student’s *t*-test results.

**Table 7 antibiotics-12-00177-t007:** (**A**–**D**). Ames test (*Salmonella typhimurium*, strain TA98 and TA100) with and without metabolic activation (S9 mix) to assay mutagen protective activity of the four studied EOs (0.01–1.0 mg/plate concentration range) in presence of direct-acting mutagen 2-nitrofluorene (2 mg/plate; NF) and sodium azide (1 mg/plate; SA) and of the indirectly acting mutagen 2-aminoanthracene (2 mg/plate; AA). Negative controls (EOs = 0.000 mg/plate) were set up with 100 mL/plate of DMSO. The maximum concentration value (1.0 mg/plate) of the concentration range coincides with the maximum dose in which the EOs do not express cytotoxicity in HUD evaluation (Table 6A–D). Table 7A: *Zingiber officinale* EO mutagen-protective property; Table 7B: *Curcuma longa* mutagen-protective property; Table 7C: *Ocimum campechianum* mutagen-protective property; Table 7D: *Cymbopogon citratus* mutagen-protective property. The results are expressed both as revertants percentage (Rev.%) ± standard deviation (s.d.) and Colony Forming Units (CFU)/plate ± standard deviation.

A				
*Z. officinale* (mg/plate)	TA98-S9	TA98+S9 AA	TA100-S9 SA	TA100+S9 AA
0.00	506.0 ± 65.9	1218.0 ± 101.6	1188.0 ± 128.4	581.0 ± 68.7
0.01	498.0 ± 56.4	1231.0 ± 76.8	1212.0 ± 76.5	592.0 ± 56.3
0.02	512.0 ± 81.2	1287.0 ± 74.3	1203.0 ± 59.7	567.0 ± 44.7
0.05	538.0 ± 53.4	1183.0 ± 81.4	1225.0 ± 65.0	543.0 ± 38.7
0.10	410.0 ± 26.4	1114.0 ± 52.3	1116.0 ± 73.1	534.0 ± 33.3
0.20	185.0 ± 22.2 *	1036.0 ± 38.5	1197.0 ± 29.9	519.0 ± 27.4
0.50	28.0 ± 3.6 *	228.0 ± 11.0 *	821.0 ± 44.7 *	233.0 ± 6.5 *
1.00	- *	- *	- *	161.0 ± 4.8 *
**B**				
** *C. longa* ** **(mag/plate)**	**TA98-S9 NF**	**TA98+S9 AA**	**TA100-S9 SA**	**TA100+S9 AA**
0.00	363.0 ± 28.4	1233.0 ± 74.4	1179.0 ± 99.4	508.0 ± 33.6
0.01	380.0 ± 22.6	1208.0 ± 68.7	1103.0 ± 81.5	489.0 ± 48.7
0.02	361.0 ± 30.7	1177.0 ± 59.5	1121.0 ± 60.0	515.0 ± 36.4
0.05	352.0 ± 17.5	1165.0 ± 66.3	1136.0 ± 55.7	553.0 ± 30.4
0.10	227.0 ± 11.8 *	974.0 ± 52.1	1148.0 ± 68.4	421.0 ± 12.5 *
0.20	191.0 ± 9.9 *	370.0 ± 27.8 *	1154.0 ± 61.7	192.0 ± 13.7 *
0.50	35.0 ± 2.3 *	57.0 ± 3.8 *	825.0 ± 31.4 *	37.0 ± 2.4 *
1.00	- *	- *	266.0 ± 21.0 *	- *
**C**				
** *O. campechianum* ** **(mg/plate)**	**TA98-S9 NF**	**TA98+S9 AA**	**TA100-S9 SA**	**TA100+S9 AA**
0.00	431.0 ± 38.1	643.0 ± 42.8	1034.0 ± 77.6	889.0 ± 71.5
0.01	397.0 ± 27.5	592.0 ± 38.6	1006.0 ± 69.4	881.0 ± 68.2
0.02	202.0 ± 31.4 *	388.0 ± 27.9 *	1015.0 ± 55.8	827.0 ± 59.7
0.05	189.0 ± 22.9 *	247.0 ± 17.5 *	1069.0 ± 48.1	712.0 ± 63.4
0.10	168.0 ± 12.8 *	210.0 ± 12.4 *	1063.0 ± 53.7	406.0 ± 28.6 *
0.20	64.0 ± 3.4 *	78.0 ± 4.4 *	560.0 ± 32.6 *	287.0 ± 7.8 *
0.50	- *	- *	592.0 ± 40.1 *	85.0 ± 5.0 *
1.00	- *	- *	101.0 ± 22.0 *	- *
**D**				
** *C. citratus* ** **(mg/plate)**	**TA98-S9 NF**	**TA98+S9 AA**	**TA100-S9 SA**	**TA100+S9 AA**
0.00	201.0 ± 17.6	377.0 ± 26.7	787.0 ± 68.5	683.0 ± 50.7
0.01	198.0 ± 15.4	371.0 ± 22.3	788.0 ± 71.4	655.0 ± 39.4
0.02	199.0 ± 16.8	389.0 ± 18.0	795.0 ± 58.3	679.0v42.6
0.05	202.0 ± 13.6	385.0 ± 19.2	808.0 ± 62.7	677.0 ± 28.6
0.10	194.0 ± 7.9	276.0 ± 11.4 *	814.0 ± 55.5	560.0 ± 31.5 *
0.20	81.0 ± 6.1 *	83.0 ± 7.6 *	698.0 ± 42.0	375.0 ± 11.3 *
0.50	- *	- *	372.0 ± 12.8 *	173.0 ± 10.7 *
1.00	- *	- *	80.0 ± 6.6 *	62.0 ± 3.8 *

(*) Significant values (Student *t*-test).

## Data Availability

The data presented in this study are available in this article.

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
