# Peer review of "A Comparative Study on Chemical Compositions and Biological Activities of Four Amazonian Ecuador Essential Oils: Curcuma longa L. (Zingiberaceae), Cymbopogon citratus (DC.) Stapf, (Poaceae), Ocimum campechianum Mill. (Lamiaceae), and Zingiber officinale Roscoe (Zingiberaceae)"

_antibiotics, 2023, doi:10.3390/antibiotics12010177_

Round 1

Reviewer 1 Report

The MS entitled “A comparative study on chemical compositions and biological activities of four Amazonian Ecuador essential oils: Curcuma longa L. (Zingiberaceae), Cymbopogon citratus (DC.) Stapf, (Poaceae), Ocimum campechianum Mill. (Lamiaceae), and Zingiber officinale Roscoe (Zingiberaceae)” was thoroughly reviewed. The MS is technically correct and has been well formulated. Some minor corrections are suggested. My concerns/suggestions in the MS are provided as under:

1. In abstract, GC-FID, GC-MS, and HS-GC-FID-MS should be written in full at first appearance.

2. Page 1, line 33, in abstract: DPPH should be corrected.

3. page 2, introduction, line 50. Correct the word “World”.

4. In introduction. Add some recent literature.

 Khan, F. A., Khan, N. M., Ahmad, S., Aziz, R., Ullah, I., Almehmadi, M., ... & Aljuaid, A. (2022). Phytochemical profiling, antioxidant, antimicrobial and cholinesterase inhibitory effects of essential oils isolated from the leaves of Artemisia scoparia and Artemisia absinthium. Pharmaceuticals15(10), 1221.

5. Page 21, line 600. Write the equation in MS equation properly. 

6.The figure 3 should be corrected for better display. Use separate figures for each sampel.

7. The obtained of Headspace from essential oils is not satisfactory. Could it be possible?

8. Add control in antioxidant assay. What type of control was used since the authors used tween 40 in these assays. Was there any standard drug used in DPPH assay? If not, state the reason in experimental section, page 19, line 500.

Author Response

Reply to the Reviewer #1 

Comments and Suggestions for Authors

  1. In abstract, GC-FID, GC-MS, and HS-GC-FID-MS should be written in full at first appearance.

The acronyms have been reported in full

  1. Page 1, line 33, in abstract: DPPH should be corrected.

To reduce the abstract, the definitions of DPPH and ABTS were removed.

  1. page 2, introduction, line 50. Correct the word “World”.

The word World has been rewritten in lowercase as suggested

  1. In introduction. Add some recent literature.

The following reference has been added:

Khan, F.A.; Khan, N.M.; Ahmad, S.; Aziz, R.; Ullah, I.; Almehmadi, M.; Allahyani, M.; Alsaiari, A.A.; Aljuaid, A. Phytochemical profiling, antioxidant, antimicrobial and cholinesterase inhibitory effects of essential oils isolated from the leaves of Artemisia scoparia and Artemisia absinthium. Pharmaceuticals 2022, 15(10), 1221. DOI: 10.3390/ph15101221

  1. Page 21, line 600. Write the equation in MS equation properly. 

The equation was amended as requested

  1. The figure 3 should be corrected for better display. Use separate figures for each sample.

Figure 3 was modified as requested

  1. The obtained of Headspace from essential oils is not satisfactory. Could it be possible?

If by "Headspace” you mean the vapours released by the essential oil in the agar vapour assay, the results shown are those acquired, supported by a method already established in our laboratory and in past and present literature (for e.g., Ovidi E, Laghezza Masci V, Zambelli M, Tiezzi A, Vitalini S, Garzoli S. Laurus nobilis, Salvia sclarea and Salvia officinalis Essential Oils and Hydrolates: Evaluation of Liquid and Vapor Phase Chemical Composition and Biological Activities. Plants. 2021; 10(4):707. https://doi.org/10.3390/plants10040707; Santomauro F, Donato R, Pini G, Sacco C, Ascrizzi R, Bilia AR. Liquid and Vapor-Phase Activity of Artemisia annua Essential Oil against Pathogenic Malassezia spp. Planta Medica. 2018; 84(3):160-167. doi:10.1055/s-0043-118912).

Our experiments were carried out at different concentrations and showed a good dose-response correlation.

  1. Add control in antioxidant assay. What type of control was used since the authors used tween 40 in these assays. Was there any standard drug used in DPPH assay? If not, state the reason in experimental section, page 19, line 500.
    The material and method paragraph about antioxidant activity has been rephrased to highlight positive and negative controls used in both the assays.

Finally, the English language throughout the manuscript was improved.

Reviewer 2 Report

Comments and Suggestions for Authors

The manuscript submitted for review is focused on chemical  characterization of Essential oils (EOs) from cultivated plants of Curcuma longa (Zingiberaceae), Cymbopogon citratus (Poaceae), Ocimum campechianum (Lamiaceae), and Zingiber officinale (Zingiberaceae) grown in Amazonian Ecuador area, one of the largest biodiverse hot spots in the World. Aromatic plants have always characterized the ethnobotanical traditions and cultures of all societies in the World without distinction, finding a place as an economic botany resource in many uses such as healthy preparations and therapeutic applications, cosmetics, perfuming and sanitizing environments, insect repellent, for religious and pagan propitiatory rites. Therefore, it is of particular interest to verify the effectiveness of traditional uses from a phytochemical and bioactive point of view and to evaluate any new uses related to the peculiarities of the chemical composition due to the specificity of the cultivation area. The objective need to use standardized EOs in terms of quality and quantity of constituents explains the need for supplies from cultivated rather than spontaneous aromatic species, to be able to control the environment and cultivation practices by minimizing the variability of the composition of the phytocomplexes.

The chemical composition of the EOs and of the volatile fraction (headspace, HS) have been performed, together with the evaluation and comparison of the antioxidant activity and antimicrobial properties related to identifying the compounds mainly responsible for the bioactivity. Finally, the mutagen-protective capacity through the Ames test was verified and compared among all the four Amazonian EOs, with the aim of suggesting, considering all the biological activities performed, a possible functional and protective role of essential oils in applicative and health-related fields such as for example the food sector.

The structure of the manuscript is in accordance with the requirements of “Antibiotics”. The research methods applied are appropriate, comprehensive and sufficient to achieve the objectives of the study. The illustrative material (tables and figures) are representatives and of good quality.

However, some comments need to be made:

Abstract

It should be shortened - exceed the recommended 200 words

Materials and Methods

Line 432: Fresh plant parts from three different stocks of … - clarify what is meant by stocks:  variety, locality?

Line 620: the sentence need redaction, as follows: Relative standard deviations and statistical significance were evaluated using Student's t-test; (P < 0.05)

Results and Discussion

Line 118: “ Ecuadorian essential oils (EOs)…, it should be “The essential oils (EOs)…,

Table 2.: The title should be:  Chemical composition of essential oils (EOs) from studied species through GC and GC-MS analyses. Inside the table, the species names should be written in full, e.g.: Curcuma longa, Cymbopogon citratus, Ocimum campechianum, Zingiber officinale. The legend for: a, b, c, d: “aCompounds are listed in order of elution from a Varian FactorFour VF-5 ms col-170 umn. bAI (arithmetic retention index) calculated on a Varian FactorFour VF-5 ms column. ctr = trace (< 0.1%). dDash indicates not detected.” to be given as a footnote below the table. The same goes for the title of Table 3.

Table 4: The title should be:  Antioxidant properties of EOs from studied species through DPPH (the transcription of the abbreviation is presented in the text, it is unnecessary here) and ABTS (the transcription of the abbreviation is presented in the text, it is unnecessary here) tests in comparison with that of commercial Thymus vulgaris (phytocomplex as positive control), and Trolox® (pure chemical compound as positive control). The legend for : IC50= half maximal concentration (IC50) to determine the antioxidant  effect” to be given as a footnote below the table.

Table 5: The title should be:  Antimicrobial activities of  Eos from studied species compared with that of commercial T. vulgaris. Chloramphenicol (CPh) and fluconazole (Flu) were used to test the sensitivity of bacterial and yeast strains respectively. The legend for: MIC: Minimum Inhibitory Concentration, considered as the lowest concentration of each DMSO/EO solution deposited on the sterile paper disc showing a clear zone of inhibition” to be given as a footnote below the table.

Table 6: “(-) no Colony Forming Units (CFU) has been detected because of the cytotoxicity expressed by the essential oils (EOs). (*) Significant evidence (cytotoxicity) in light of Student’s t-test results.” should be given as a footnote below the table.

Table 7: “(*) Significant values (Student t test)” should be given as a footnote below the table.

Figure captions should be given after the figures.

Figure 2: In the caption of the figure the description of figures a: and b: is missing

In all text “Amazonian Eos” to be change with “Eos from studied species”, “Amazonian Eos species”, “Eos from Amazonian species”

The “evidence”( the emerged evidence - line 662) or “evidenced” (the EOs (HS) evidenced C. longa EO – Abstract, line 23, The antioxidant activity of the EOs evidenced O. campechianum phytocomplex as…- line 638) to be changed with more appropriate term according to the context : e.g. the results in first case, and revealed, in the second case, and at all check the accuracy of the terms used.

In conclusion, this manuscript is recommended for publication in “Antibiotics”.

Author Response

Reply to the Reviewer #2

Comments and Suggestions for Authors

The manuscript submitted for review is focused on chemical  characterization of Essential oils (EOs) from cultivated plants of Curcuma longa (Zingiberaceae), Cymbopogon citratus (Poaceae), Ocimum campechianum (Lamiaceae), and Zingiber officinale (Zingiberaceae) grown in Amazonian Ecuador area, one of the largest biodiverse hot spots in the World. Aromatic plants have always characterized the ethnobotanical traditions and cultures of all societies in the World without distinction, finding a place as an economic botany resource in many uses such as healthy preparations and therapeutic applications, cosmetics, perfuming and sanitizing environments, insect repellent, for religious and pagan propitiatory rites. Therefore, it is of particular interest to verify the effectiveness of traditional uses from a phytochemical and bioactive point of view and to evaluate any new uses related to the peculiarities of the chemical composition due to the specificity of the cultivation area. The objective need to use standardized EOs in terms of quality and quantity of constituents explains the need for supplies from cultivated rather than spontaneous aromatic species, to be able to control the environment and cultivation practices by minimizing the variability of the composition of the phytocomplexes.

The chemical composition of the EOs and of the volatile fraction (headspace, HS) have been performed, together with the evaluation and comparison of the antioxidant activity and antimicrobial properties related to identifying the compounds mainly responsible for the bioactivity. Finally, the mutagen-protective capacity through the Ames test was verified and compared among all the four Amazonian EOs, with the aim of suggesting, considering all the biological activities performed, a possible functional and protective role of essential oils in applicative and health-related fields such as for example the food sector.

The structure of the manuscript is in accordance with the requirements of “Antibiotics”. The research methods applied are appropriate, comprehensive and sufficient to achieve the objectives of the study. The illustrative material (tables and figures) are representatives and of good quality.

However, some comments need to be made:

Abstract

It should be shortened - exceed the recommended 200 words

The abstract has been shortened

Materials and Methods

Line 432: Fresh plant parts from three different stocks of … - clarify what is meant by „stocks“:  variety, locality?

The sentence has been clarified in §3.1.

Line 620: the sentence need redaction, as follows: Relative standard deviations and statistical significance were evaluated using Student's t-test; (P < 0.05)

The sentence has been redacted.

Results and Discussion

Line 118: “Ecuadorian essential oils (EOs)…, it should be “The essential oils (EOs)…,

The sentence has been changed as suggested.

Table 2.: The title should be:  Chemical composition of essential oils (EOs) from studied species through GC and GC-MS analyses. Inside the table, the species names should be written in full, e.g.: Curcuma longa, Cymbopogon citratus, Ocimum campechianum, Zingiber officinale. The legend for: a, b, c, d: “aCompounds are listed in order of elution from a Varian FactorFour VF-5 ms col-170 umn. bAI (arithmetic retention index) calculated on a Varian FactorFour VF-5 ms column. ctr = trace (< 0.1%). dDash indicates not detected.” to be given as a footnote below the table. The same goes for the title of Table 3.

Tables 2 and 3 have been changed as suggested.

Table 4: The title should be:  Antioxidant properties of EOs from studied species through DPPH (the transcription of the abbreviation is presented in the text, it is unnecessary here) and ABTS (the transcription of the abbreviation is presented in the text, it is unnecessary here) tests in comparison with that of commercial Thymus vulgaris (phytocomplex as positive control), and Trolox® (pure chemical compound as positive control). The legend for : IC50= half maximal concentration (IC50) to determine the antioxidant  effect” to be given as a footnote below the table.

Table 4 has been changed as suggested.

Table 5: The title should be:  Antimicrobial activities of  Eos from studied species compared with that of commercial T. vulgaris. Chloramphenicol (CPh) and fluconazole (Flu) were used to test the sensitivity of bacterial and yeast strains respectively. The legend for: MIC: Minimum Inhibitory Concentration, considered as the lowest concentration of each DMSO/EO solution deposited on the sterile paper disc showing a clear zone of inhibition” to be given as a footnote below the table.

Table 5 has been changed as suggested.

Table 6: “(-) no Colony Forming Units (CFU) has been detected because of the cytotoxicity expressed by the essential oils (EOs). (*) Significant evidence (cytotoxicity) in light of Student’s t-test results.” should be given as a footnote below the table.

Table 6 has been changed as suggested.

Table 7: “(*) Significant values (Student t test)” should be given as a footnote below the table.

Table 4 has been changed as suggested.

Figure captions should be given after the figures.

The captions have been moved after the figures.

Figure 2: In the caption of the figure the description of figures a: and b: is missing

The description of figures a and b have been added.

In all text “Amazonian Eos” to be change with “Eos from studied species”, “Amazonian Eos species”, “Eos from Amazonian species”

The sentence has been changed as suggested throughout the manuscript

The “evidence”( the emerged evidence - line 662) or “evidenced” (the EOs (HS) evidenced C. longa EO – Abstract, line 23, The antioxidant activity of the EOs evidenced O. campechianum phytocomplex as…- line 638) to be changed with more appropriate term according to the context : e.g. the results in first case, and revealed, in the second case, and at all check the accuracy of the terms used.

The English language has been revised throughout the text and more appropriate terms according to the context have been used